# The Role of Liquid Biopsy in the Diagnostic Testing Algorithm for Advanced Lung Cancer

Aaron C. Tan [1,2]

1   Division of Medical Oncology, National Cancer Centre Singapore, Singapore 169610, Singapore; aaron.tan@singhealth.com.sg; Tel.: +65-6436-8000; Fax: +65-6225-6283
2   Duke-NUS Medical School, National University of Singapore, Singapore 169857, Singapore

**Simple Summary:** The utility, feasibility, and cost-effectiveness of upfront tissue next-generation sequencing (NGS) for advanced non-small cell lung cancer (NSCLC) have been previously demonstrated. However, significant limitations with tissue NGS remain such as insufficient tissue, scheduling limitations, the need for repeat biopsies, and long turnaround times. This leads to delays in commencing appropriate first-line therapy, particularly in patients without a targetable alteration. Liquid biopsies, using plasma circulating tumor DNA (ctDNA), have the potential to overcome these issues. This paper puts forth the rationale for liquid biopsy as a complementary diagnostic assay in the diagnostic algorithm for advanced lung cancer.

**Abstract:** The discovery of therapeutically targetable oncogenic driver alterations has led to marked improvements in NSCLC outcomes. Targeted agents have been approved for an expanding list of biomarkers. Consequently, the accurate and timely identification of targetable alterations with diagnostic molecular profiling is crucial. The use of multiplexed tissue assays, such as next-generation sequencing (NGS), has increased significantly. However, significant limitations with tissue NGS remain, such as insufficient tissue, scheduling limitations, the need for repeat biopsies, and long turnaround times. Liquid biopsies, using plasma circulating tumor DNA (ctDNA), have the potential to overcome these issues, with simpler sample processing requirements, greater convenience, and better patient acceptability. In particular, an early liquid biopsy may allow patients access to highly effective therapies faster, allow better symptom control and quality of life, prevent rapid clinical deterioration, and reduce patient anxiety at diagnosis. More broadly, it may also allow for the more cost-effective delivery of healthcare to patients.

**Keywords:** circulating tumor DNA; ctDNA; liquid biopsy; lung cancer; non-small cell lung cancer

## 1. Introduction

Lung cancer remains amongst the most commonly diagnosed cancers and is still a leading cause of cancer death worldwide [1,2]. However, with the emergence of precision oncology with targeted therapies and the era of immunotherapy with immune checkpoint inhibitors in recent decades, survival outcomes in advanced lung cancer have substantially improved [3]. In particular, targeted therapies are approved for an expanding list of biomarkers, including *BRAF*, *EGFR*, *HER2*, and *KRAS* mutations; *MET* alterations; and *ALK*, *NTRK*, *RET,* and *ROS1* fusions [4]. Consequently, the accurate and timely identification of targetable alterations with molecular profiling to guide initial treatment decision-making in advanced non-small cell lung cancer (NSCLC) is crucial. Furthermore, there can be significant geographic diversity in the prevalence and incidence of oncogenic driver alterations. Most notably, for *EGFR* mutations, there is a much higher prevalence in Asian NSCLC populations, at around 40–60%, compared with Western populations (10–15%) [5]. As a result, in Asian populations, the proportion of patients harboring a targetable molecular

alteration may even be as high as 70% [6]. This further highlights that molecular profiling in NSCLC is paramount.

Traditionally, single-biomarker or single-gene testing has been performed routinely with techniques such as immunohistochemistry (IHC), fluorescence in situ hybridization (FISH), polymerase chain reaction (PCR), and Sanger sequencing [7]. Indeed, in many centers and countries, this remains standard practice. More recently, however, multiplex testing with techniques such as next-generation sequencing (NGS) has gained traction and allows for the evaluation of multiple biomarkers in a single workflow. The utility, feasibility, and cost-effectiveness of upfront tissue NGS in newly diagnosed advanced NSCLC have been previously demonstrated [6,8]. However, significant limitations with tissue NGS remain, such as insufficient tissue, biopsy scheduling limitations, the need for repeat biopsies, and long turnaround times [9]. Tumor heterogeneity and risks of biopsies in patients with underlying co-morbidities are further limitations of tissue NGS. This commonly leads to substantial delays in commencing appropriate first-line therapy. This even includes patients without a targetable alteration, in whom the absence of such an alteration remains vital in determining standard upfront therapy. Moreover, patients with advanced NSCLC are often highly symptomatic and can rapidly deteriorate in condition [10]—and the need to initiate therapy early without a molecular profiling result is frequently encountered. With changes in the treatment landscape of NSCLC and the incorporation of immune checkpoint inhibitor therapy in the first-line setting [11], this may potentially expose patients to ineffective and costly therapies that are also associated with significant toxicities.

Liquid biopsies, using plasma circulating tumor DNA (ctDNA) NGS (Figure 1), have the potential to overcome many limitations of tissue-based NGS testing [12]. In particular, there is often a much faster turnaround time compared to tissue NGS, predominantly due to simpler and fewer requirements in sample processing. From a patient perspective, it is minimally invasive and provides greater convenience and acceptability. Liquid biopsies may be obtained from a range of fluid sources, including blood, urine, saliva, and effusions—and a range of analytes, such as cell-free DNA/RNA, circulating tumor cells, and extracellular vesicles. However, currently, liquid biopsies using plasma ctDNA analysis are the most advanced in the field of oncology. Plasma ctDNA genotyping has been demonstrated to have high analytical specificity and positive-predictive value (PPV) when compared with tissue-based testing [13–17], and the feasibility of using liquid biopsies has been confirmed in numerous settings [17–19]. Notably, several liquid biopsy assays are now approved as companion diagnostics for approved targeted therapies [20–23], facilitating the increased uptake of liquid biopsy in standard clinical practice. Lower sensitivity, however, can be seen with liquid biopsy due to low levels of detectable ctDNA [24]. Therefore, currently, plasma ctDNA genotyping is considered a complementary assay to tissue-based testing. Nevertheless, there is growing acceptance of a 'plasma first' approach to biomarker evaluation, as evidenced in international consensus statements [25].

Greater innovation in the diagnostic algorithm for advanced NSCLC with liquid biopsy may also relate to the timing of plasma collection and testing (Figure 2). For example, a pilot study has explored the role of early liquid biopsy at the time of tissue biopsy to expedite the initiation of first-line therapy [26]. In patients with high disease and symptom burden, the early initiation of therapy may allow for better symptom control and quality of life, provide early access to therapy to prevent rapid deterioration, and reduce patient anxiety. In addition, reducing the need for tissue-based testing, fewer repeat tissue biopsies, and limiting the initiation of other less-effective systemic therapies are other potential benefits. In patients with non-oncogene addicted NSCLC, there is also increasing evidence for the potential role of liquid biopsies as a predictive biomarker in monitoring for response and resistance to therapy [27]. Identifying the most cost-effective interventions in the diagnostic molecular profiling of advanced NSCLC, however, requires formal cost-effectiveness studies to fully understand the optimal approach within different healthcare systems.

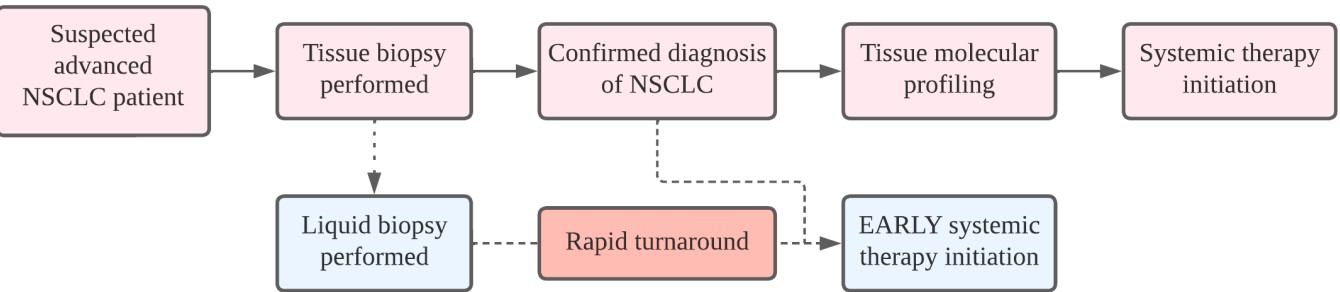

**Figure 1.** Liquid biopsy for non-small cell lung cancer (NSCLC) using plasma circulating tumor DNA (ctDNA).

**Figure 2.** Potential diagnostic testing algorithm for advanced non-small cell lung cancer (NSCLC) incorporating liquid biopsy.

While the formal histopathological diagnosis of lung cancer and PD-L1 IHC is still firmly entrenched as standard components of the diagnostic testing paradigm, whether liquid biopsy could replace tissue biopsy in the future remains an open question. With recent approvals of targeted therapies for tissue-agnostic biomarkers [28], such as *NTRK* gene fusions, the appropriate initiation of systemic therapy in these cases (albeit rare) without a formal histological diagnosis could be argued. Alternatively, in cases where attempts at tissue acquisition may be difficult or non-yielding, the identification in plasma of an alteration considered pathognomonic for lung cancer (such as the 'classical' activating *EGFR* mutations) [29]—could conceivably obviate the need for histological tissue diagnosis confirmation. Finally, there is the potential for circulating or plasma biomarkers, such as methylation markers, which may even detect or predict tumor histology [30]. This foreshadows the potential role of liquid biopsy in early stage lung cancer, whether as a screening or diagnostic assay or to stratify patients for curative therapies—although there is limited evidence in the space to date [31].

## 2. Conclusions

The treatment landscape for lung cancer has evolved substantially over the past decade in terms of rapid drug development, particularly with targeted therapies. With an increasing proportion of patients with targetable oncogenic driver alterations, the timely identification of these molecular alterations is especially paramount. Harnessing techno­logical advancements with liquid biopsies provides a unique opportunity to transform the

diagnostic testing paradigm. In addition, with an increased uptake of molecular tumor boards, therapeutic decisions according to individual patients' genotypes can be optimized. Formal cost-effectiveness analyses, standardization across assays, and implementation studies are still needed to fully understand the role of liquid biopsy in guiding first-line therapeutic decisions. Ultimately, however, an early liquid biopsy may allow patients access to highly effective therapies faster, allow better symptom control and quality of life, prevent rapid clinical deterioration, and reduce patient anxiety at diagnosis. It may also have the potential to translate into improved treatment and survival outcomes and allow for more efficient healthcare delivery to patients.

**Funding:** This research received no external funding.

**Institutional Review Board Statement:** Not applicable.

**Informed Consent Statement:** Not applicable.

**Data Availability Statement:** Not applicable.

**Conflicts of Interest:** A.C.T. reports consultant or advisory roles for Amgen and Pfizer outside of the submitted work.

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
