# Peer review of "The Role of Liquid Biopsy in the Diagnostic Testing Algorithm for Advanced Lung Cancer"

_onco, doi:10.3390/onco2030012_

Round 1
Reviewer 1 Report
The author presented a commentary about the emerging role of liquid biopsy in the diagnostic testing paradigm for advanced lung cancer.
This article is very interesting and well-written.
I agree to publish it in the present form. Only a small suggestion (not mandatory) for lines 56-58: it should be better to avoid the repetition of "targetable alteration".
Reviewer 2 Report
In this commentary, the author offers an overview of liquid biopsy from a clinical point. Generally, until a few years ago, liquid biopsy using plasma circulating tumour DNA (ctDNA), has moved from single-gene analysis (i.e. EGFR mutations) to guideline-recommended broad-based plasma ctDNA analysis by NGS. So the shift towards major technical advances has dramatically changed the application of a therapeutic algorithm for patients with advanced NSCLC. Although the commentary appears to be well thought through, I do have a few questions that require clarification, which might necessitate minor changes to the manuscript text. My comments/suggestions below are in no-particular order.
1) First of all, I would ask you to clarify the concept of liquid biopsy, since throughout the text you just mentioned plasma-derived circulating tumor DNA (ctDNA), but actually there are several other biological sources (urine, cerebro-spinal fluid, blood, saliva, effusions, and analytes, including circulating tumor cells, ctRNA, and extracellular vesicles). Please clarify this point in the introduction.
2) In particular, MTB is emerging as a useful tool to solve complex cases and improve patients’ treatment outcomes: basically, an optimal therapeutic decision has done for each patient according to their genomic profile. In this context, how could be important to translate into improved treatment and survival outcomes and allow for more efficient healthcare delivery to NSCLC patients? Please explain in the conclusion.
3) Ultimately, for patients with oncogene-addicted NSCLC progressing after a targeted therapy, liquid biopsy can reveal the tumor clonal evolution and provide useful information to drive subsequent therapeutic choices. I would like to ask you what the role is for ctDNA in patients with nononcogene-addicted NSCLC? Traces the differences between them in the rationale.
4) One of the main disadvantages of liquid biopsy is related to the standardization of the crucial pre-analytical and analytical phases to reduce to a minimum the risk of false results. It would be important that you integrate and emphasize this fact in supporting treatment-decision making through the exchange of practice-based insights on how to best solve doubtful and complex cases. I would suggest adding in the conclusion section.
5) With the continued development of more powerful and sensitive assays, these techniques will empower clinicians to better characterize early-stage disease and can be used in the screening of high-risk patients, which may eliminate the requirement for tissue diagnosis in some settings. How much it could help us to stratify early NSCLC patients? Discuss it.
Reviewer 3 Report
In this journal commentary, the author A.C. Tan, outlines the role of liquid biopsy in advanced NSCLC and the role of multiplex of profiling of ctDNA over tissue genotyping by NGS. With increasing identification of new biomarkers and corresponding targeted therapies in the treatment of advanced NSCLC patients, the commentary highlights key advantages of the use of ctDNA profiling from liquid biopsies and how this non-invasive approach holds clinical potential in the treatment of NSCLC patients by overcoming delays in treatment initiation, avoidance of unnecessary toxicities and improved responses and outcomes in patients. Moreover, it also addresses the potential cost-effectiveness of using liquid biopsy thereby reducing healthcare costs in the long-term.
Some minor corrections to current commentary:
[1] Line 12, replace "...including in patients..." with "...particularly in patients..."
[2] Line 13, replace "has" with "have".
[3] In the section outlining some of the limitations with tissue NGS (lines 53-54), perhaps the author could include additional recognized limitations such as tumor heterogeneity and patients with underlying co-morbidities?
[4] In lines 71-74, additional references should be included for other FDA-approved liquid biopsies as companion diagnostics such as the FoundationOne Liquid CDx and the Cobas EGFR mutation v2 assay?
[5] Line 82, being sentence with "In patients with a high disease..." and remove "Especially" at the beginning of the sentence.
[6] Line 106, remove the word "especially" from this sentence.
